# Effect of Dietary Guanidinoacetic Acid Levels on the Mitigation of Greenhouse Gas Production and the Rumen Fermentation Profile of Alfalfa-Based Diets

**DOI:** 10.3390/ani13111719

**Published:** 2023-05-23

**Authors:** Oscar Vicente Vazquez-Mendoza, Veronica Andrade-Yucailla, Mona Mohamed Mohamed Yasseen Elghandour, Diego Armando Masaquiza-Moposita, Jorge Adalberto Cayetano-De-Jesús, Edwin Rafael Alvarado-Ramírez, Moyosore Joseph Adegbeye, Marcos Barros-Rodríguez, Abdelfattah Zeidan Mohamed Salem

**Affiliations:** 1Facultad de Medicina Veterinaria y Zootecnia, Universidad Autónoma del Estado de México, Toluca 50295, Mexico; osvam2009@gmail.com (O.V.V.-M.); mmohamede@uaemex.mx (M.M.M.Y.E.); adal_cay@hotmail.com (J.A.C.-D.-J.); 2Centro de Investigaciones Agropecuarias, Facultad de Ciencias Agrarias, Universidad Estatal Península de Santa Elena, La Libertad 240204, Ecuador; crisita_2725@hotmail.com; 3Facultad de Ciencias Pecuarias, Escuela Superior Politécnica de Chimborazo, Sede Orellana, El Coca 220150, Ecuador; dmasaquiza@espoch.edu.ec; 4Unidad Académica Multidisciplinaria Mante, Universidad Autónoma de Tamaulipas, El Mante 89840, Mexico; win.rmz@hotmail.com; 5Department of Animal Production and Health, Federal University of Technology, Akure 340110, Nigeria; alanspeco@yahoo.com; 6Facultad de Ciencias Agropecuarias, Universidad Técnica de Ambato, Cevallos 1801334, Ecuador; ma_barrosr@yahoo.es

**Keywords:** alfalfa, carbon monoxide, enteric methane, guanidinoacetic acid, hydrogen sulfide, rumen fermentation

## Abstract

**Simple Summary:**

Alfalfa (*Medicago sativa* L.) is considered the queen of forages, and hay is an important source of protein and fiber for livestock, while guanidinoacetic acid (GAA) is a feed additive that can improve growth performance and energy metabolism in animals and reduce the population of methanogenic microorganisms. However, the percentage of alfalfa hay (AH) in the diet can cause variations in greenhouse gas (GHG) production, the rumen fermentation profile and methane (CH_4_) conversion efficiency, which, in turn, influences the effectiveness of GAA. In this regard, this study demonstrates that the percentage of AH in the diet affects the effectiveness of GAA and that the addition of GAA in diets with 25 and 100% AH presents low effectiveness, a diet with 10% AH can improve the mitigation of GHG and the rumen fermentation profile without compromising the CH_4_ conversion efficiency using a dose of 0.0015 or 0.0020 g GAA g^−1^ DM in the diet.

**Abstract:**

The objective of this study was to evaluate the effect of different percentages of alfalfa (*Medicago sativa* L.) hay (AH) and doses of guanidinoacetic acid (GAA) in the diet on the mitigation of greenhouse gas production, the in vitro rumen fermentation profile and methane (CH_4_) conversion efficiency. AH percentages were defined for the diets of beef and dairy cattle, as well as under grazing conditions (10 (AH10), 25 (AH25) and 100% (AH100)), while the GAA doses were 0 (control), 0.0005, 0.0010, 0.0015, 0.0020, 0.0025 and 0.0030 g g^−1^ DM diet. With an increased dose of GAA, the total gas production (GP) and methane (CH_4_) increased (*p* = 0.0439) in the AH10 diet, while in AH25 diet, no effect was observed (*p* = 0.1311), and in AH100, GP and CH_4_ levels decreased (*p* = 0.0113). In addition, the increase in GAA decreased (*p* = 0.0042) the proportion of CH_4_ in the AH25 diet, with no influence (*p* = 0.1050) on CH_4_ in the AH10 and AH100 diet groups. Carbon monoxide production decreased (*p* = 0.0227) in the AH100 diet with most GAA doses, and the other diets did not show an effect (*p* = 0.0617) on carbon monoxide, while the production of hydrogen sulfide decreased (*p* = 0.0441) in the AH10 and AH100 diets with the addition of GAA, with no effect observed in association with the AH25 diet (*p* = 0.3162). The pH level increased (*p* < 0.0001) and dry matter degradation (DMD) decreased (*p* < 0.0001) when AH was increased from 10 to 25%, while 25 to 100% AH contents had the opposite effect. In addition, with an increased GAA dose, only the pH in the AH100 diet increased (*p* = 0.0142 and *p* = 0.0023) the DMD in the AH10 diet group. Similarly, GAA influenced (*p* = 0.0002) SCFA, ME and CH_4_ conversion efficiency but only in the AH10 diet group. In this diet group, it was observed that with an increased dose of GAA, SCFA and ME increased (*p* = 0.0002), while CH_4_ per unit of OM decreased (*p* = 0.0002) only with doses of 0.0010, 0.0015 and 0.0020 g, with no effect on CH_4_ per unit of SCFA and ME (*p* = 0.1790 and *p* = 0.1343). In conclusion, the positive effects of GAA depend on the percentage of AH, and diets with 25 and 100% AH showed very little improvement with the addition of GAA, while the diet with 10% AH presented the best results.

## 1. Introduction

In recent years, the warming of the Earth has increased as a consequence of the increase in the concentration of greenhouse gases (GHGs) in the atmosphere, which has also caused instability in environmental conditions, especially in rainfall and atmospheric temperature [1]. According to the FAO, livestock farming is responsible for around 18% of methane (CH_4_) emissions and 9% of carbon dioxide (CO_2_) production [2], as these gases are the result of ruminal fermentation of feed, mainly from fibrous carbohydrates; and although their production is inevitable, high amounts represent a loss of gross energy of between 2 and 12% for the animals [3]. Other gases produced by rumen include carbon monoxide (CO), which indirectly contributes to global warming as a precursor to ozone, and hydrogen sulfide (H_2_S) [4], can provide an alternate sink for metabolic hydrogen (H_2_), which decreases CH_4_ production [5]. However, H_2_S is easily absorbed in the intestinal wall of ruminants, so in high concentrations, it can be toxic and even induce polioencephalomalacia, a harmful brain disease in animals [6]. Consequently, the need has arisen to propose novel and rapid strategies for the mitigation of GHGs of animal origin [7].

High-quality and digestible feed has been considered as an option to reduce GHG emissions by enabling increased production of short-chain fatty acids (SFCA) and an increase the energy supply in animals [8], which would be reflected in higher animal performance. In this sense, alfalfa (*Medicago sativa* L.), also called the “queen of forages”, is a perennial legume that is widely cultivated [9] and is considered a source of protein for ruminants [10], since it has a high protein content with a rapid degradation rate and a high proportion of rapidly degrading protein [11]. As a forage crop, alfalfa is not only rich in protein but also in vitamins and some minerals, with high palatability, making it a viable alternative to be used in ruminant diets [12,13]. In this regard, it has been reported that alfalfa can improve the characteristics and quality of the carcass in sheep and even equal or exceed the quality of sheep fed with concentrate, since the animals fed with alfalfa have presented with a high concentration of linoleic acid, which has benefits for human health [14,15]. Other studies reported that the inclusion of alfalfa hay (AH) in lamb feed delayed lipid oxidation and myoglobin formation in meat, thereby prolonging meat shelf life [16]. In addition, alfalfa presents secondary metabolites such as phenols, flavonoids and saponins [17]; therefore, it also represents an important source of bioactive compounds [18]. However, forages preserved by tedding have lower protein digestibility compared to their fresh or ensiled form [19], and these changes may influence the abundance of the microbial population, which, in turn, affects feed kinetics, digestibility, GHG emissions and fermentation end products [20].

The use of feed additives has also been considered as an alternative, with improved growth performance of animals, GHG emissions per unit of animal product decrease. Guanidinoacetic acid (GAA) is an additive that is naturally biosynthesized from arginine and glycine in the kidneys or pancreas of vertebrates and is a direct precursor of creatine biosynthesis [21,22], which is why it participates in energy metabolism and is used in animal feed, including that of ruminants [23]. In bulls, it has been reported that GAA increases the microbial population and improves the rumen fermentation profile and nutrient digestibility without altering blood biochemistry parameters [8,24]. Meanwhile, in sheep, performance, carcass characteristics and meat nutritional content have been reported to improve in association with GAA [25,26]. Other studies reported that GAA in ruminants increased daily weight gain, feed conversion efficiency and total short-chain fatty acid (SCFA) production [27]. However, the doses of GAA vary across studies, and the reported effects show variability. Therefore, considering that the GAA doses used in the previous studies were tested with a diet containing a forage source other than AH and given the benefits of AH for ruminant nutrition, we hypothesized that the percentage of AH in the diet can affect the level of response of GAA to the parameters of rumen digestibility and the final products of fermentation. Therefore, the objective of this study was to evaluate the effect of different percentages of AH and doses of GAA in the diet on the mitigation of the production of greenhouse gases (CH_4_, CO and H_2_S), the in vitro ruminal fermentation profile and the CH_4_ conversion efficiency.

## 2. Materials and Methods

### 2.1. Experimental Treatments

The study factors were the percentage of alfalfa (*Medicago sativa* L.) hay (AH) and the dose of guanidinoacetic acid (GAA) in the diet of ruminant livestock. The percentage of inclusion of AH in the diet was defined for beef cattle fattening and dairy cattle under grazing conditions as 10, 25 and 100% (only AH), respectively. The doses of GAA were based on previous studies in ruminants: 0.0000 (control), 0.0005, 0.0010, 0.0015, 0.0020, 0.0025 and 0.0030 g GAA g^−1^ DM diet, for which they were evaluated in a total of 21 treatments. 

### 2.2. Diets and Chemical Composition

The AH was obtained from a local business in the municipality of Toluca, State of Mexico, Mexico, and according to the information provided by the seller, the alfalfa was second cut and was harvested in the full-bloom stage at 5 cm above ground level. GAA was purchased from Evonik México S.A. de C.V. under the trade name of GuanAMINO^®^ for ruminants, and at the time of evaluation, it had a purity of 96%. Once the diets were prepared, representative samples were dehydrated at 60 °C for 72 h and crushed in a hammer mill (Thomas Wiley^®^ Laboratory Mill model 4, Swedesboro, NJ, USA) with a 1 mm sieve, and the chemical composition was determined. The analysis included the estimation of dry matter (#934.01), ash (#942.05), nitrogen (#954.01) and ethereal extract (#920.39) according to the description of the AOAC [28], while the determinations of neutral detergent fiber and acid detergent fiber were performed using an ANKOM^200^ fiber analyzer (ANKOM Technology Corp, Macedon, NY, USA) with alpha-amylase and sodium sulfite according to the methodology proposed by Van Soest et al. [29]. Organic matter was estimated (g kg^−1^ DM) by subtracting the ash content from 1000, while protein was estimated by multiplying the nitrogen content by 6.25. Table 1 shows the ingredients and the chemical composition of the experimental diets formulated with the different percentages of AH.

### 2.3. In Vitro Fermentation

#### 2.3.1. Collection of Inoculum and Preparation of the Nutrient Medium

The inoculum was obtained from four steers (420 ± 20 kg LW) slaughtered in the municipal slaughterhouse of Toluca, State of Mexico, Mexico, and for the collection of the rumen content and the transfer to the laboratory, hermetic thermoses were used. The extraction of the ruminal liquid was carried out by filtering the rumen content with four layers of cheesecloth, which was then mixed and kept at 39 °C until use. The nutrient medium contained buffer solution, macrominerals, microminerals, resazurin and distilled water and was prepared following methodology described by Goering and Van Soest [30]. Before incubation, the ruminal inoculum and the nutrient medium were mixed at a ratio of 1:4 (*v*/*v*) using a magnetic stirrer, with the temperature maintained at 39 °C. 

#### 2.3.2. Incubation Process

Prior to incubation, 500 mg of each diet was weighed and placed in a glass flask with a capacity of 160 mL; then, the doses of GAA and 50 mL of the obtained solution of ruminal inoculum and the nutrient medium were added. The bottles were sealed with butyl rubber stoppers and aluminum seals, shaken and incubated in an incubator at 39 °C for the evaluation period, which, in this case, corresponded to 48 h. Each treatment was incubated in triplicate, and three flasks without substrate were added as blanks to correct for gas measurements.

#### 2.3.3. Measurement of Gas Production

The volume of the total gas production (GP) was measured after 2, 4, 12, 24, 27, 30 and 48 h of incubation, following the technique proposed by Theodorou et al. [31] using a digital manometer (Manometer model 407910, Extech^®^ Instruments, Nashue, NH, USA). Methane (CH_4_), carbon monoxide (CO) and hydrogen sulfide (H_2_S) were quantified at the same time points as the total gas but with a portable gas detector (Dräger X-am^®^, model 2500, Dräger, Lübeck, SH, Germany) equipped with an external pump (Dräger X-am^®^, Dräger, Lübeck, SH, Germany) in which a known amount of gas was injected, simultaneously indicating the percentage of each gas [32]. At the end of each measurement, the gas accumulated in the headspace of the vials was released to avoid partial dissolution of the gases and erroneous estimates [33].

#### 2.3.4. pH and Dry Matter Degradation

At the end of the incubation, the contents of the vials were filtered following the methodology described by Alvarado-Ramírez et al. [34], which consisted of retaining the residual substrate in bags with 25 µm porosity (Filter bags F57, ANKOM Technology Corp, Macedon, NY, USA) and collecting the liquid in beakers. The pH of the liquids was measured using a potentiometer with a glass electrode (pH wireless electrode HALO^®^ model HI11102, Hanna^®^ Instruments, Woonsocket, RI, USA), while the residual substrate was washed with plenty of water and dehydrated at 60 °C for 48 h to calculate the dry matter degradation by weight difference.

#### 2.3.5. Calculations

The kinetics of production of GP, CH_4_, CO and H_2_S were estimated by adjusting the volume of the gases with the NLIN procedure of SAS [35] according to the model proposed by France et al. [36]:
y = *b* × [1 − e^−*c*(t−*Lag*)^](1)
where:y = volume (mL) of GP, CH_4_, CO and H_2_S at time *t* (h);*b* = asymptotic GP, CH_4_, CO and H_2_S production (mL g^−1^ DM);*c* = the rate GP, CH_4_, CO and H_2_S production (mL h^−1^);*Lag* = the initial delay time before the beginning of GP, CH_4_, CO and H_2_S production (h).

Metabolizable energy (ME; MJ kg^−1^ DM) was estimated according to the equation proposed by Menke et al. [37]:ME = 2.20 + (0.136 × GP) + (0.057 × CP)(2)
where:GP = total gas production (mL 200 mg^−1^ DM) after 24 h of incubation;CP = crude protein (g kg^−1^ DM).

Short-chain fatty acid (SCFA; mmol 200 mg^−1^ DM) concentrations were calculated according to Getachew et al. [38]:SCFA = (0.0222 × GP) − 0.00425(3)
where GP = total gas production (mL 200 mg^−1^ DM) after 24 h of incubation.

Additionally, the ratios of CH_4_ to SCFA (CH4:SCFA; mmol mmol^−1^), ME (CH4:ME; g MJ^−1^) and OM (CH4:OM; mL g^−1^) were calculated.

### 2.4. Statistical Analysis

The experimental design was completely randomized with a 3 × 7 bifactorial arrangement; a factor A corresponded to the percentage of AH, and factor B corresponded to the GAA doses, with three repetitions per treatment. The analysis was carried out with the GLM procedure of the SAS program [35] according to the following statistical model:*Y_ijk_* = µ + *A_i_* + *B_j_* + (*A* × *B*)*_ij_* + ε*_ijk_*(4)
where Y*_ijk_* is the response variable, μ is the overall mean, *A_i_* is the effect of the percentage of AH, *B_j_* is the effect of the dose of GAA, (*A* × *B*)*_ij_* is the effect of the interaction between the percentage of AH and the dose of GAA and ε*_ijk_* is the experimental error. Linear and quadratic polynomial contrasts were used to evaluate the response of the different percentages of AH with increasing doses of GAA in the diet. Tukey’s test was applied for comparison of means, with significant differences considered when *p* < 0.05.

## 3. Results

### 3.1. In Vitro Ruminal Total Gas (GP) Production

Figure 1 shows the effect of the different percentages of alfalfa hay (AH) and doses of guanidinoacetic acid (GAA) in the diet on the in vitro rumen production kinetics of GP. With increased AH contents from 10 (AH10) to 25% (AH25) in the diet, the asymptotic production (mL g^−1^ DM) and the production rate (mL GP h^−1^) of GP increased (*p* < 0.0001), whereas when AH contents were increased from 25 to 100% (AH100) asymptotic production and the GP production rate decreased, and the time (h) in the delay phase presented an opposite effect (*p* = 0.0094). At 4 and 24 h, GP production (mL GP g^−1^ DM incubated and degraded) decreased (*p* < 0.0001) with augmentation of AH content, while at 48 h, it increased (*p* < 0.0001) with augmentation of AH content from 10 to 25%, then decreased when AH content was increased from 25 to 100% (Figure 1A; Table 2). GAA doses did not affect (*p* = 0.8443) asymptotic production but decreased (*p* = 0.0343) the rate of GP production and increased (*p* = 0.0255) the time in the lag phase. At 24 h, the production of GP (mL GP g^−1^ DM incubated and degraded) increased (*p* < 0.0001) with the addition of GAA, except with doses of 0.0005 and 0.0010 g, with which GP production decreased (Figure 1B; Table 2). However, with respect to the interaction (*p* = 0.0330) between the percentage of AH and the GAA dose, it was observed that GAA only affected the parameters of GP production in the AH100 diet, and although the GAA doses did not show a trend, all or most decreased (*p* = 0.0135) asymptotic production and the total gas production rate and increased (*p* = 0.0081) the time in the lag phase. In addition, it was observed that with respect to the production of GP, there was interaction (*p* = 0.0010) at 24 and 48 h but only (*p* = 0.0405) in the AH10 diet, while in the AH25 diet, there was no interaction (*p* = 0.2780), and in the AH100 diet, an interaction was only observed (*p* = 0.0013) at 48 h. In the AH10 diet, GP production (mL GP g^−1^ DM incubated and degraded) increased with increasing GAA dose both at 24 and 48 h, except with the 0.0005 g dose, while in the AH10 diet, AH100 decreased with increasing dose (Table 2). 

### 3.2. In Vitro Ruminal Methane (CH_4_) Production

Figure 2 shows the effect of the different percentages of AH and the dose of GAA in the diet on the kinetics of in vitro rumen production of CH_4_. The asymptotic production (mL CH_4_ g^−1^ DM) and the production rate (mL CH_4_ h^−1^) of CH_4_ increased (*p* = 0.0038) when AH contents in the diet were increased from 10 to 25%, while with an increase in AH contents from 25 to 100%, they decreased. At 4 and 24 h, CH_4_ production decreased (*p* ≤ 0.0001) as the AH percentage increased, while at 48 h CH_4_ production first increased, then decreased (*p* < 0.0001). However, in the case of the CH_4_ proportion (mL CH_4_ 100 mL^−1^ GP), there was no significant decrease at 24 h (Figure 2A; Table 3). In contrast, GAA did not affect (*p* = 0.2799) the parameters of CH_4_ production, and after 4 h of incubation, all doses of GAA increased (*p* = 0.0349) the production and proportion of CH_4_ (mL CH_4_ 100 mL^− 1^ GP), except for 0.0005 and 0.0010 g, for which the production and proportion of CH_4_ decreased (Figure 2B; Table 3). However, the interaction between the percentage of AH and the dose of GAA had an effect (*p* = 0.0438) on the asymptotic production of CH_4_ and the production of CH_4_ throughout incubation. In the AH10 diet, asymptotic production increased (*p* = 0.0103) with the addition of GAA, with no influence in the AH25 diet group (*p* = 0.0965), and in AH100 it decreased (*p* = 0.0113). In addition, GAA increased (*p* = 0.0439) the production of CH_4_ at 4 and 48 h in the AH10 diet, with no effect in the AH25 diet group, (*p* = 0.1244) and in the AH100 diet group, it only decreased CH_4_ production (*p* = 0.0113) at 48 h. However, GAA increased (*p* = 0.0041) the proportion of CH_4_ at 4 h in the AH10 diet and decreased CH_4_ production at 48 h in the AH25 diet (*p* = 0.0042), with no influence in the AH100 diet group (*p* = 0.1050; Table 3).

### 3.3. In Vitro Ruminal Carbon Monoxide (CO) Production

Figure 3 shows the effect of the different percentages of AH and the doses of GAA in the diet on the kinetics of in vitro rumen production of CO. The asymptotic production (mL CO g^−1^ DM) and the production rate (mL CO h^−1^) of CO indicate an effect (*p* = 0.0005) of the percentage of AH, as both increased when the percentage of AH was increased from 10 to 25%, while when AH content was increased from 25 to 100%, they both decreased. At 4 h, CO production decreased (*p* < 0.0001) as the percentage of AH increased, while at 24 and 48 h, CO production first increased (*p* < 0.0001), then decreased (Figure 1A; Table 4). Instead, the doses of GAA did not influence (*p* = 0.4968) the production parameters but did influence the production of CO after 24 h of incubation, at which point an increase in the dose of GAA increased (*p* = 0.0084) the production of CO, except with doses of 0.0005 and 0.0010 g, for which CO production decreased (Figure 3B; Table 4). However, an interaction (*p* = 0.0413) was observed between the percentage of AH and the dose of GAA for the rate and production of CO throughout the incubation period. With an increase in the dose of GAA, the rate of CO production increased (*p* = 0.0086) in the AH10 diet, except with doses of 0.0005 and 0.0010 g, while in the AH25 and AH100 diets, no effect was observed (*p* = 0.4457). Furthermore, with most GAA doses, CO production increased (*p* = 0.0397) at 4 h in the AH10 diet and at 24 h in the AH25 diet, while in the AH100 diet, CO production only decreased (*p* = 0.0227) at 48 h (Table 4).

### 3.4. In Vitro Ruminal Hydrogen Sulfide (H_2_S) Production

Figure 4 shows the effect of the different percentages of AH and the doses of GAA in the diet on the kinetics of in vitro rumen production of H_2_S. The percentage of AH did not affect (*p* = 0.0724) the parameters, but it did (*p* = 0.0136) affect the production of H_2_S; when the percentage of AH was increased from 10 to 25%, the production of H_2_S decreased (*p* = 0.0136), while when AH content was increased from 25 to 100%, H_2_S production increased, and the AH100 diet surpassed (*p* = 0.0136) the AH10 in the production of H_2_S after 48 h (Figure 4A; Table 5). Similarly, GAA doses did not influence (*p* = 0.4699) the parameters but did (*p* = 0.0034) influence the H_2_S production throughout the incubation period. Although no trend was observed, H_2_S production decreased (*p* = 0.0034) with the addition of GAA throughout the incubation period, except with a dose of 0.0030 g at 48 h, for which H_2_S production increased (Figure 4B; Table 5). In this case, the interaction between the AH percentage and the GAA dose was significant (*p* = 0.0032) with respect to H_2_S production throughout the incubation period; when the GAA dose was increased, the H_2_S production in the diet AH10 decreased (*p* < 0.0001) throughout incubation, and in the AH100 diet, it increased (*p* = 0.0141) at 48 h with most doses (Table 5).

### 3.5. In Vitro Rumen Fermentation Profile and CH_4_ Conversion Efficiency

The percentage of AH influenced (*p* = 0.0009) the ruminal fermentation profile and the efficiency of CH_4_ production, except in CH_4_ per unit of short-chain fatty acids (SCFA). It was observed that the pH first increased (*p* < 0.0001) with an increase in AH from 10 to 25%; then, when AH was increased from 25 to 100%, the pH decreased. On the other hand, the degradation of dry matter (DMD), the SCFA, metabolizable energy (ME) and CH_4_ per unit of ME and organic matter (OM) presented an opposite effect (*p* = 0.0009). Like the percentage of AH, the dose of GAA influenced (*p* = 0.0151) the rumen fermentation profile and CH_4_ per unit of OM, and although no trend was observed when increasing the dose, the addition of GAA decreased (*p* = 0.0011) pH and increased (*p* = 0.0151) DMD, SCFA, ME and the CH_4_ per unit of OM unit with most doses. However, the SCFA, the ME and the CH_4_ conversion efficiency presented an effect (*p* = 0.0438) on the interaction between the percentage of AH and the dose of GAA. It was observed that only the AH10 diet presented an effect (*p* = 0.0002) of GAA on these variables. In this diet, SCFA and ME increased (*p* = 0.0002) with increasing GAA dose, except with doses of 0.0005 and 0.0010 g, while the amount of CH_4_ per OM increased (*p* = 0.0002) with doses of 0.0005, 0.0025 and 0.0030 g and decreased with doses of 0.0010, 0.0015 and 0.0020 g (Table 6).

## 4. Discussion

### 4.1. In Vitro Ruminal Total Gas Production 

The total gas production (GP) during fermentation is positively correlated with the degradability of feed nutrients [39], and the degree of degradability is determined by the accessibility of feed components for rumen microorganisms, the activity of rumen microbes and the time available for fermentation [40]. That is why the total gas production and the rate of gas production are used as indicators to assess the degradability of feed and the functionality and adaptability of rumen microbes to the diet [41]. In this study, after 24 h of fermentation, the total gas production was higher in the diets with 10 (AH10) and 25% (AH25) alfalfa hay (AH) compared to the diet containing 100% hay (AH100), while after 48 h, the AH10 and AH100 diet presented similar levels of gas production, which were lower than the total gas obtained with the AH25 diet. This can be attributed to the content of easily fermentable carbohydrates in each diet, as they provide energy to the ruminal microbiota for their metabolic activities during the first hours of fermentation [42], as was observed after 24 h in the AH10 and AH25 diets, which presented a high total gas production. In contrast, the AH100 diet showed fewer fast-fermenting carbohydrates and higher total gas production up to 48 h, indicating that rumen microbes took time to adapt to the diet and had less energy available for their offspring activities compared to the other diets [43]. 

However, it has been reported that the addition of GAA in the diet can decrease the asymptotic production and the production rate of GP [44], which was observed in the AH100 diet when increasing the dose of GAA, which is attributed to the proportion of short-chain fatty acids (SCFA), since GAA favors the formation of propionate [45], an SCFA that produces less gas compared to acetate and butyrate [46]. In addition, the *lag* phase also increased in this diet; although it did not show a trend with an increasing dose of GAA, this increase can be attributed to the time that ruminal microorganisms require to adapt to the presence of GAA, since it is susceptible to degradability when it is not rumen protected [47], and microorganisms can use it as a source of energy and nitrogen to synthesize their proteins [45]. However, with an increase in GAA, GP production increased in the AH10 diet, while in the AH100 diet, GP production decreased with increased GAA dose. In the case of the AH10 diet, this result can be attributed to the fact that GAA increases the activity of fibrolytic enzymes, α-amylase and protease [45], as well as the populations of total bacteria, fungi, *Ruminococcus albus*, *Fibrobacter succinogenes*, *Ruminococcus flavefaciens*, *Ruminobacter amilophilus* and *Prevotella ruminicola* [26,48], which favors the degradation of the diet, resulting in a consequent increase in GP. In contrast, in the AH100 diet, GAA possibly did not favor enzymatic activity and an increase in the populations of fibrolytic bacteria since GAA did not influence dry matter degradation (DMD) in this diet. In addition, as previously mentioned, it is possible that increasing the dose of GAA favored the formation of propionate, which leads to a lower production of GP, as supported by the lower proportion of methane.

### 4.2. In Vitro Ruminal Methane (CH_4_) Production

The production of CH_4_ in the AH25 diet can be attributed to the higher total gas production that it presented, since feeds that present high total gas production generally also show a higher production of gas CH_4_ [41] because in the rumen, the main biochemical process carried out by bacteria, protozoa and fungi is the fermentation of carbohydrates, and as a byproduct, they release short-chain fatty acids (SCFA; mainly acetic, butyric and propionic acids), metabolic hydrogen (H_2_) and carbon dioxide (CO_2_) [49]. Subsequently, methanogenic archaea remove H_2_ by reducing CO_2_ to CH_4_ [42,50] to maintain low H_2_ concentrations in the rumen [51], since, otherwise, inhibition of microbial growth and feed degradation would occur [52]. This metabolic process is known as methanogenesis [53] and is influenced by the fiber content in the food, since fibrous feeds tend to produce more H_2_ and therefore more CH_4_ [2]. Despite the influence of fiber, the AH100 diet did not produce the most CH_4_, which is attributed to the bioactive compounds that this plant presents, since they have antimicrobial and protozoan properties that reduce CH_4_ production [54]. In addition, it is important to note that the inclusion of GAA in the AH25 and AH100 diets decreased the production and proportion of CH_4_ with most of the doses, which is attributed to the decrease exerted by GAA on the population of the protozoa and methanogens [26,48] and because propionate production increases, which decreases the acetate:propionate ratio and the metabolic H_2_ available for the formation of CH_4_ [23,26]. However, in the AH10 diet, the results were inconsistent because the inclusion of GAA increased CH_4_ production but did not affect the proportion of CH_4_ with respect to total gas production. On the other hand, the ruminal population of sulfo-reducing bacteria (SRB) was low but had the ability to compete with methanogens for H_2_ for the production of H_2_S [55], although it is likely that the inclusion of GAA in the diet caused an inhibitory effect on these bacteria, and therefore, this metabolic pathway did not function as an alternate H_2_ sink. In addition, the availability of sulfur (S) is necessary to increase the relative abundance of SRBs and their ability to compete with methanogens for H_2_ [56].

### 4.3. In Vitro Ruminal Carbon Monoxide Production

Carbon monoxide (CO) is a metabolic intermediate gas that, under anaerobic conditions, is produced by anaerobic microbes during the degradation of organic matter (MO) [57]. In the rumen, which is also an anaerobic environment, in addition to the amount of degraded OM, other factors influence the production of CO, including microbial activity and the fermentative capacity of the ruminal microbiota, as well as the type of degraded chemical components (fiber, protein, lipids, etc.) [34]. In addition, some CO dehydrogenase enzymes are highly dependent on the availability of trace minerals, are capable of reducing CO_2_ and oxidizing CO by utilizing their catalytic groups for electron transfer and can maintain redox homeostasis during digestion of the feed [58]. Therefore, the variations in the production of CO are attributed to the degraded components of each diet, since although they were similar in protein content, they differed in terms of the other chemical and mineral components (Table 1). In turn, the variations in the chemical composition influence the populations of rumen microorganisms, such as methanogens, acetogens and SRB, which require CO for their metabolism [59]. Therefore, the high production of CO in the AH25 and AH100 diets is attributed to a greater availability of substrates for these microorganisms, and the production of CO increased even more with the dose of GAA, as it results in partial degradation in the rumen and is used by some microorganisms as a substrate for their metabolic functions [60]. In addition, it was observed that the production of CO was positively related to the production of CH_4_ of each diet, which may indicate that methanogens maintain a synergy with other microorganisms that produce CO, since in the presence of water, acetogens and methanogens oxidize CO to form CO_2_ and H_2_ and subsequently produce CH_4_ [61,62].

### 4.4. In Vitro Ruminal Hydrogen Sulfide (H_2_S) Production

It has been reported that most of the dietary S ingested by ruminants through feed is converted to sulfate by rumen microorganisms, especially bacteria [50,57]. The same occurs with amino acids that contain S; they are fermented to sulfate [63], which is used together with lactate as a substrate by SRB to produce sulfide, which is combined with H_2_ to form hydrogen sulfide (H_2_S) [58,64]. In this study, the S in the diets was not quantified, but the highest H_2_S production without the inclusion of GAA was observed in the AH10 diet, while the lowest H_2_S production was observed in the AH100 diet; therefore, it can be assumed that another ingredient in the diets likely contributed S, causing variations in the production of H_2_S. This assertion is supported by the metabolic process of H_2_S production described above and by the findings reported by Smith et al. [65], who reported that increasing the amount of S in the diet from 2 to 8 g kg^−1^ DM increased H_2_S production by more than 470%. In addition, there is a negative correlation between pH and H_2_S production that is attributed to the protonation of aqueous sulfide [66], which inhibits the activity of SRBs [67]. Therefore, the production of H_2_S requires an acidic rumen environment [6], preferably with a pH in a range of 5.5 to 6.5 [68]. In this study pH was in the range of 7.22 ± 0.26. (average ± standard deviation), so it cannot be ruled out that the low production of H_2_S was a consequence of the pH level. On the other hand, regardless of the amount of S available, in the three diets, most of the GAA inclusion doses presented a lower H_2_S production compared to the 0 g g^−1^ DM diet dose, and although the ratio is not known with certainty, it seems that the inclusion of GAA caused an inhibitory effect or an unfavorable environment for the activity of SRBs.

### 4.5. In Vitro Rumen Fermentation Profile and CH_4_ Conversion Efficiency

The rumen fermentation profile can be used as a measure or indicator of rumen health, feed digestibility and digestion efficiency. Rumen pH is vital for the persistence and stability of rumen microorganisms [69], while DMD, SCFA and metabolizable energy (ME) are useful as indicators of the digestibility and energy value of the feed [41,70]. In the current study, the pH at the end of fermentation ranged between 6.83 and 7.64, values that are within the range reported in other studies [71,72], and of the three diets, the AH25 diet had the highest pH, while the AH10 diet had the lowest pH. These results agree with those reported by Nemati et al. [73], who observed that increasing the percentage of AH from 15 to 30% in the diet increased the pH to a higher level than that associated with the control treatment. This is attributed to the high buffering capacity and the low content of water-soluble carbohydrates that alfalfa presents, which makes it difficult to lower the pH [74]. In addition, during the deamination of proteins, ammonia is produced, which provides additional buffering to the rumen to maintain a relatively constant pH [75]. However, the DMD, the SFCA and the ME were negatively affected by the increase in the percentage of hay, so the AH25 diet obtained the lowest values, and the diet containing 10% AH was associated with the highest values, whereas the 100% hay diet presented intermediate values. Considering that the diet influences the rumen pH and, in turn, the digestion and metabolism of nutrients [76], it is possible that the low DMD in the AH25 diet is due to the high pH level, which reduced the degradation capacity of ruminal microbes [6]. In addition, alfalfa presents a high concentration of phenolic compounds with antimicrobial bioactivity [19], which is another possible reason for the low DMD in this diet. On the other hand, the diet with 10% hay presented higher values than the AH25 diet, which is attributed to the lower percentage of alfalfa. In the case of the AH100 diet, although it presented a DMD higher than that obtained in the AH25 diet, it is important to mention that in this diet, there was no other ingredient, so the high DMD is attributed to the structure of fibrous carbohydrates. In all diets, DMD enhanced the production of SFCA and ME, since both are end products of rumen fiber fermentation [77] and positively correlated with DMD [9]. On the other hand, it has been reported that the inclusion of GAA in the diet increases the rumen microbial population and the digestibility of nutrients [26,48], which consequently influence SCFA and ME. Therefore, the high fermentation profile can be attributed to an improved rumen microbial population and nutrient digestibility, which influenced total SCFAs. In addition, the improvement in SCFA and ME evidences the role played by GAA in energy metabolism, which leads to greater energy efficiency [24]; although it seems that the conversion efficiency of CH_4_ in the AH10 diet decreased, it is important to highlight that it showed higher DMD and therefore increased the production of gases and the final products of fermentation, including CH_4_.

## 5. Conclusions

It is concluded that the positive effects of the addition of GAA depend on the percentage of AH in the diet and that the diets with 25 and 100% AH showed very little improvement with the addition of GAA, possibly representing an unnecessary expense. In contrast, the diet with 10% AH showed the best results with a dose equal to or greater than 0.0015 g, especially with a dose of 0.0020 g, which increased GP, CH_4_, DMD, SCFA and ME and decreased the H_2_S and CH_4_ per unit of OM without affecting CO production, the proportion of CH_4_ and the CH_4_ per unit of SCFA and ME. Therefore, an in vivo evaluation of a diet containing 10% AH with doses of 0.0015 and 0.0020 g GAA g^−1^ DM is suggested in order to validate that GAA can be used as a strategy to mitigate the production of greenhouse gases in ruminants.

## Figures and Tables

**Figure 1 animals-13-01719-f001:**
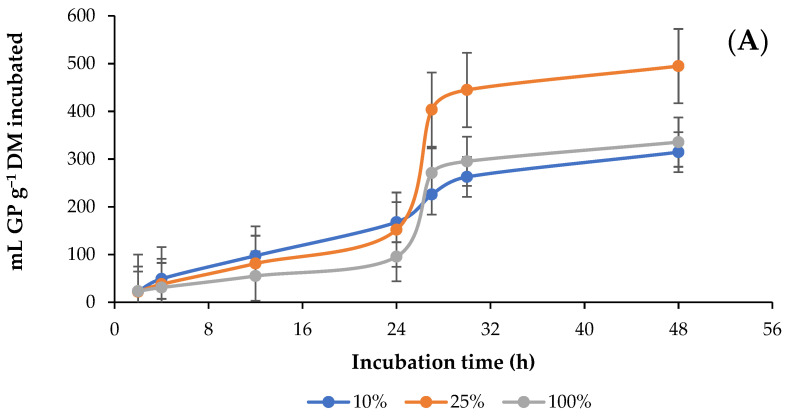
Kinetics of in vitro ruminal production of total gas production (GP) in response to different percentages of alfalfa (*Medicago sativa* L.) hay (**A**) and doses (g g^−1^ DM diet) of guanidinoacetic acid (**B**).

**Figure 2 animals-13-01719-f002:**
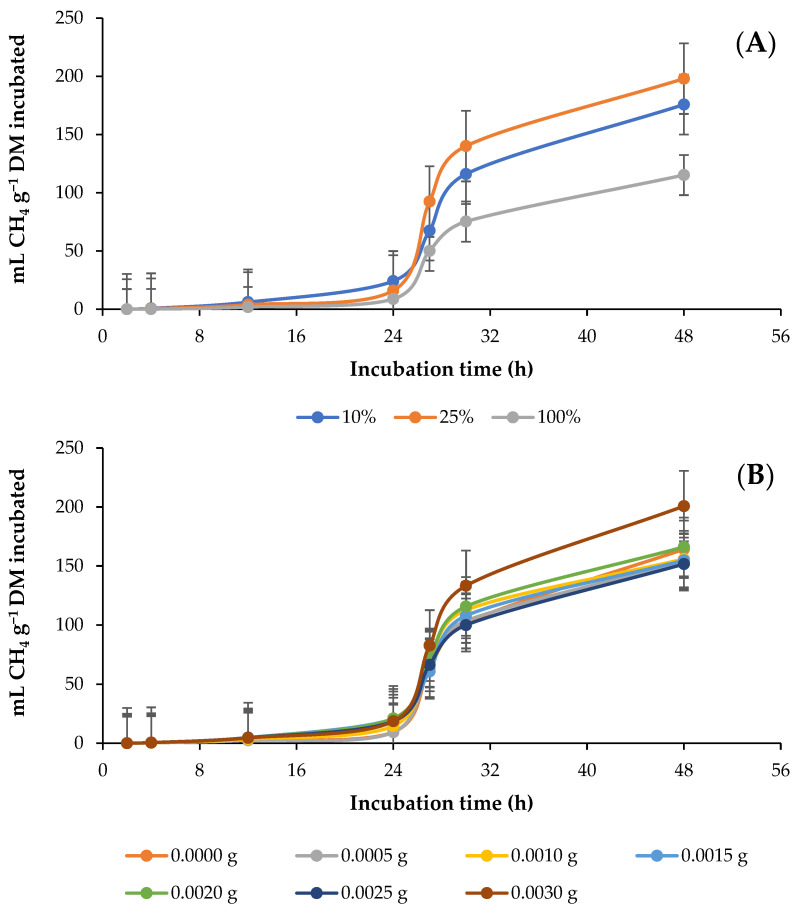
Kinetics of in vitro ruminal production of methane (CH_4_) in response to different percentages of alfalfa (*Medicago sativa* L.) hay (**A**) and doses (g g^−1^ DM diet) of guanidinoacetic acid (**B**).

**Figure 3 animals-13-01719-f003:**
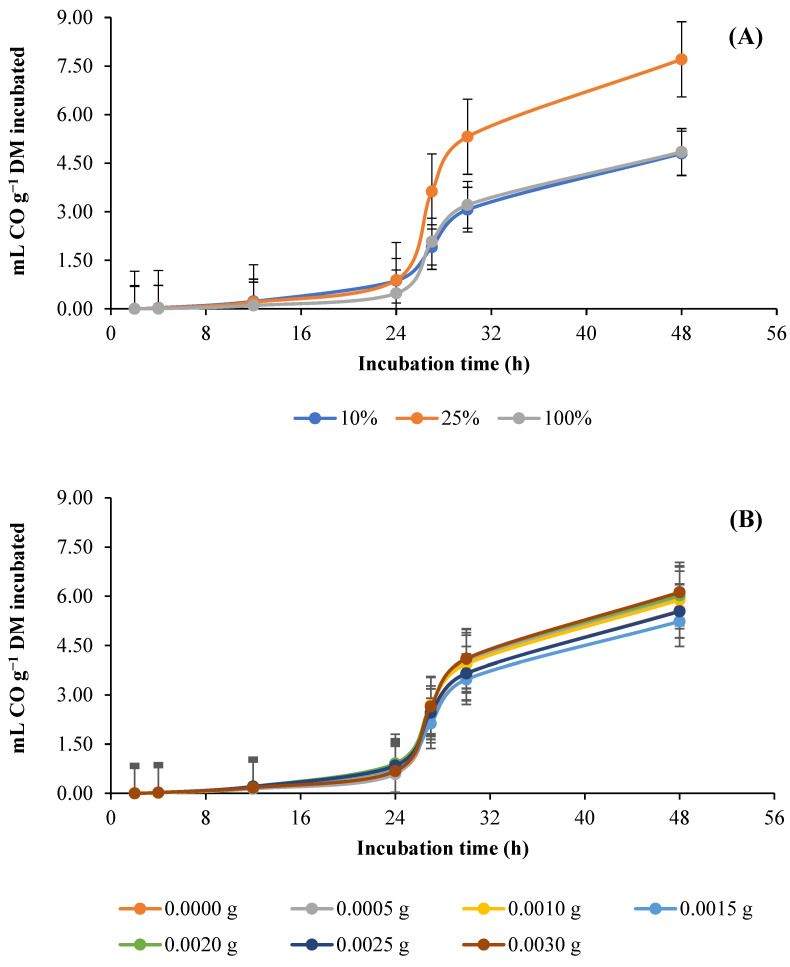
Kinetics of in vitro ruminal production of carbon monoxide (CO) in response to different percentages of alfalfa (*Medicago sativa* L.) hay (**A**) and doses (g g^−1^ DM diet) of guanidinoacetic acid (**B**).

**Figure 4 animals-13-01719-f004:**
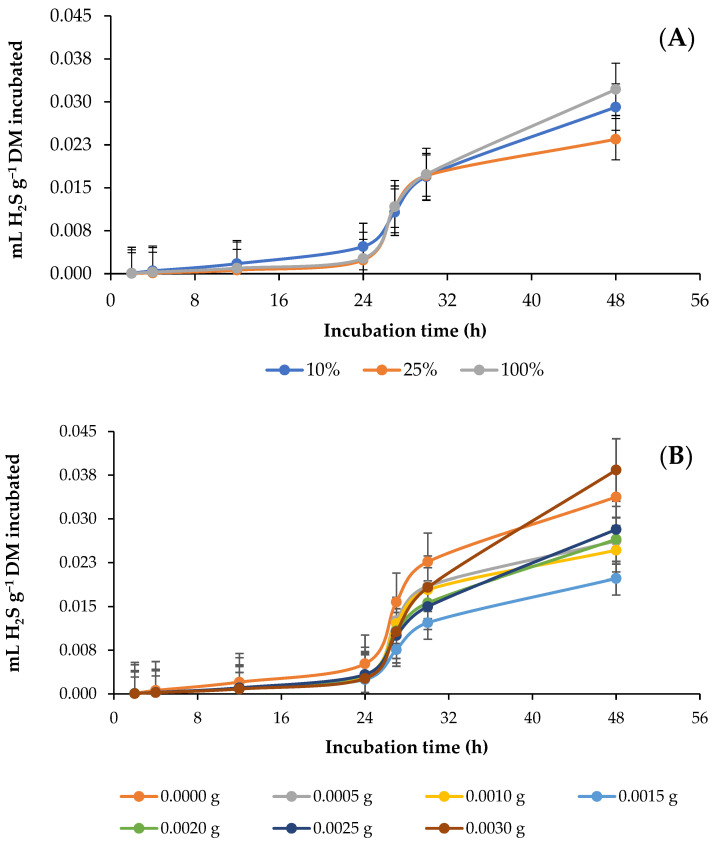
Kinetics of in vitro ruminal production of hydrogen sulfide (H_2_S) in response to different percentages of alfalfa (*Medicago sativa* L.) hay (**A**) and doses (g g^−1^ DM diet) of guanidinoacetic acid (**B**).

**Table 1 animals-13-01719-t001:** Ingredients and chemical composition of the diets formulated with different percentages of alfalfa (*Medicago sativa* L.) hay.

Item	Alfalfa Hay (%)		
	10	25	100
Ingredients			
Alfalfa hay	10.0	25.0	100.0
Wheat meal	18.0	18.0	
Maize	21.5	6.5	
Wheat bran	12.0	12.0	
Gluten feed	16.3	16.3	
Soybean meal	7.0	10.0	
Soybean husk	8.1	5.1	
Molasses	7.0	7.0	
Vit/Min	0.1	0.1	
Chemical composition			
Organic matter (g kg^−1^ DM)	946.3	942.9	969.0
Crude protein (g kg^−1^ DM)	130.9	130.7	155.0
Neutral detergent fiber (g kg^−1^ DM)	320.7	371.1	535.5
Acid detergent fiber (g kg^−1^ DM)	149.6	194.3	378.6
Ether extract (g kg^−1^ DM)	200.5	191.0	18.0

**Table 2 animals-13-01719-t002:** Parameters and in vitro ruminal total gas production (GP) in response to different percentages of alfalfa (*Medicago sativa* L.) hay and doses of guanidinoacetic acid (GAA) in the diet after 4, 24 and 48 h of incubation.

Alfalfa Hay (%)	Dose of GAA (g g^−1^ Diet)	Gas Production (GP)								
		Parameters ^1^			mL GP g^−1^ DM Incubated			mL GP g^−1^ DM Degraded		
		*b*	*c*	*Lag*	4 h	24 h	48 h	4 h	24 h	48 h
10	0.0000	302.73	0.0240	1.73	50.58	144.61	248.84	101.16	289.22	497.68
	0.0005	306.07	0.0214	3.08	50.30	121.33	241.49	100.59	242.66	482.97
	0.0010	328.60	0.0244	3.88	46.76	138.26	266.60	93.52	276.53	533.19
	0.0015	319.92	0.0306	2.98	47.86	184.90	297.99	95.71	369.82	595.99
	0.0020	388.68	0.0322	5.01	47.86	201.41	359.60	95.71	402.81	719.20
	0.0025	339.43	0.0308	3.26	50.26	191.57	316.38	100.52	383.14	632.76
	0.0030	356.75	0.1772	4.70	49.23	192.15	469.53	98.46	384.29	939.06
	SEM ^2^	32.192	0.05572	0.796	2.000	9.852	45.802	4.000	19.704	91.605
	Dose of GAA	0.5292	0.4410	0.1387	0.7553	0.0002	0.0405	0.7554	0.0002	0.0405
	Linear	0.7115	0.9341	0.2855	0.3526	0.0118	0.4606	0.3520	0.0118	0.4605
	Quadratic	0.6680	0.9673	0.1408	0.3329	0.0455	0.9049	0.3333	0.0455	0.9049
25	0.0000	415.83	0.2988	2.88	35.75	151.94	494.32	71.50	303.87	988.63
	0.0005	505.82	0.3512	3.11	34.55	146.88	523.29	69.10	293.75	1046.58
	0.0010	522.18	0.2612	2.53	39.14	151.61	521.63	78.28	303.21	1043.27
	0.0015	478.85	0.2408	5.01	40.02	155.97	477.39	80.04	311.93	954.78
	0.0020	427.05	0.1790	4.65	37.34	154.91	422.23	74.68	309.81	844.46
	0.0025	508.32	0.2092	2.70	42.49	158.41	510.65	84.98	316.82	1021.30
	0.0030	492.48	0.2865	2.94	34.97	144.03	513.29	69.94	288.05	1026.59
	SEM ^2^	41.481	0.07482	1.123	2.471	4.789	47.545	4.942	9.581	95.090
	Dose of GAA	0.4634	0.7287	0.5955	0.2780	0.3957	0.7498	0.2777	0.3959	0.7498
	Linear	0.3009	0.5922	0.2020	0.2422	0.5613	0.8049	0.2420	0.5616	0.8049
	Quadratic	0.1628	0.9266	0.3217	0.6850	0.6954	0.5488	0.6852	0.6955	0.5488
100	0.0000	399.65	0.3490	3.32	30.46	97.38	418.27	60.91	194.76	836.54
	0.0005	363.45	0.3536	3.32	28.37	91.70	381.39	56.74	183.41	762.77
	0.0010	333.15	0.3217	3.15	30.60	92.37	349.09	61.19	184.75	698.19
	0.0015	320.30	0.1150	7.20	31.44	100.32	306.87	62.89	200.65	613.74
	0.0020	390.33	0.3589	3.30	28.97	99.63	407.14	57.94	199.26	814.27
	0.0025	286.58	0.0334	8.08	32.15	94.44	256.72	64.30	188.88	513.43
	0.0030	255.53	0.0312	6.06	33.48	92.40	228.47	66.95	184.79	456.94
	SEM ^2^	26.128	0.03710	0.986	1.162	3.008	27.847	2.324	6.015	55.693
	Dose of GAA	0.0135	<0.0001	0.0081	0.0913	0.2643	0.0013	0.0913	0.2641	0.0013
	Linear	0.0498	0.0005	0.0145	0.5590	0.5008	0.0134	0.5571	0.5005	0.0134
	Quadratic	0.4160	0.0684	0.1028	0.8083	0.1004	0.6987	0.8053	0.1004	0.6987
Pooled SEM ^2^		33.861	0.05796	0.977	1.954	6.559	41.367	3.908	13.118	82.735
*p*-value										
Alfalfa hay		<0.0001	<0.0001	0.0094	<0.0001	<0.0001	<0.0001	<0.0001	<0.0001	<0.0001
Linear		<0.0001	<0.0001	0.8244	<0.0001	<0.0001	<0.0001	<0.0001	<0.0001	<0.0001
Quadratic		<0.0001	0.0145	0.0024	<0.0001	<0.0001	0.0008	<0.0001	<0.0001	0.0008
Dose of GAA		0.8443	0.0343	0.0255	0.2804	<0.0001	0.8297	0.2800	<0.0001	0.8297
Linear		0.9919	0.0509	0.0041	0.5999	0.0053	0.4390	0.5994	0.0053	0.4390
Quadratic		0.3685	0.5279	0.3407	0.7094	0.0149	0.8608	0.7086	0.0149	0.8608
Alfalfa hay × dose of GAA		0.0330	0.0019	0.0299	0.3258	<0.0001	0.0010	0.3258	<0.0001	0.0010

^1^ *b* is the asymptotic total gas production (mL GP g^−1^ DM); *c* is the rate of gas production (mL GP h^−1^); *Lag* is the initial delay before gas production begins (h). ^2^ SEM, standard error of the mean.

**Table 3 animals-13-01719-t003:** Parameters and in vitro ruminal production of methane (CH_4_) in response to different percentages of alfalfa (*Medicago sativa* L.) hay and doses of guanidinoacetic acid (GAA) in the diet after 4, 24 and 48 h of incubation.

Alfalfa Hay (%)	Dose of GAA (g g^−1^ Diet)	CH_4_ Production								
		Parameters ^1^			mL CH_4_ g^−1^ DM Incubated			mL CH_4_ 100 mL^−1^ GP		
		*b*	*c*	*Lag*	4 h	24 h	48 h	4 h	24 h	48 h
10	0.0000	87.13	0.1367	13.68	0.59	7.02	87.44	1.17	4.88	34.96
	0.0005	90.83	0.1683	13.75	0.30	7.64	91.47	0.58	6.25	36.50
	0.0010	137.87	0.1779	13.94	0.57	18.83	87.67	1.25	12.63	32.87
	0.0015	207.47	0.1423	13.08	0.80	42.98	123.30	1.67	23.25	41.75
	0.0020	225.53	0.1521	13.26	0.76	41.28	142.30	1.58	20.63	39.96
	0.0025	200.07	0.1526	13.94	1.00	25.32	116.66	2.00	13.38	37.31
	0.0030	277.02	0.1803	14.18	1.03	25.89	148.10	2.08	13.38	33.17
	SEM ^2^	34.070	0.015	0.398	0.111	9.753	14.925	0.223	5.084	4.457
	Dose of GAA	0.0103	0.3381	0.4760	0.0044	0.1072	0.0439	0.0041	0.1787	0.7559
	Linear	0.0256	0.7984	0.3020	0.2169	0.0207	0.1114	0.1347	0.0228	0.2998
	Quadratic	0.8244	0.0594	0.2686	0.3742	0.6137	0.3494	0.5510	0.8211	0.3325
25	0.0000	233.50	0.1810	14.22	0.42	13.45	234.83	1.17	8.88	47.88
	0.0005	193.18	0.2056	13.66	0.49	12.45	199.23	1.42	8.63	37.38
	0.0010	189.73	0.2325	13.79	0.44	15.27	196.60	1.17	10.00	37.42
	0.0015	174.10	0.1948	13.69	0.47	14.82	176.88	1.17	9.50	36.00
	0.0020	141.78	0.1813	13.78	0.30	16.28	142.66	0.83	10.63	33.63
	0.0025	171.00	0.1886	13.48	0.56	17.52	175.32	1.33	11.13	34.29
	0.0030	258.23	0.1619	13.67	0.46	22.03	260.50	1.25	15.25	50.75
	SEM ^2^	26.195	0.023	0.196	0.077	2.199	27.708	0.197	1.495	2.920
	Dose of GAA	0.0965	0.5113	0.2920	0.4419	0.1244	0.1311	0.5338	0.0978	0.0042
	Linear	0.1311	0.6808	0.0734	0.6552	0.6663	0.1613	1.0000	0.7713	0.0122
	Quadratic	0.6677	0.1406	0.5118	0.9862	0.6807	0.7892	1.0000	0.6637	0.2269
100	0.0000	168.55	0.1719	14.12	0.10	8.04	170.28	0.33	8.25	40.42
	0.0005	149.45	0.1632	13.42	0.00	12.68	151.87	0.00	13.75	39.59
	0.0010	132.25	0.1670	13.90	0.00	9.78	133.43	0.00	10.38	37.79
	0.0015	80.45	0.1921	14.46	0.08	4.54	80.64	0.25	4.50	26.00
	0.0020	128.27	0.1965	14.01	0.09	5.48	130.24	0.33	5.50	31.84
	0.0025	79.03	0.1317	12.98	0.06	13.04	79.65	0.17	13.75	30.25
	0.0030	60.26	0.1833	13.93	0.00	7.89	60.32	0.00	8.63	26.21
	SEM ^2^	19.639	0.013	0.279	0.045	3.084	20.029	0.144	3.158	4.129
	Dose of GAA	0.0113	0.0465	0.0406	0.4309	0.3938	0.0113	0.3809	0.3066	0.1050
	Linear	0.0068	0.2833	0.4030	0.7221	0.4353	0.0069	0.6893	0.4152	0.0270
	Quadratic	0.7521	0.3547	0.2765	0.1219	0.3705	0.7500	0.1212	0.3188	0.3803
Pooled SEM ^2^		27.280	0.018	0.303	0.083	6.041	21.538	0.191	3.562	3.892
*p*-value										
Alfalfa hay		<0.0001	0.0038	0.6843	<0.0001	0.0001	<0.0001	<0.0001	0.0872	0.0128
Linear		0.1911	0.0009	0.6823	<0.0001	0.0153	<0.0001	0.0076	0.1338	0.1601
Quadratic		<0.0001	0.6998	0.4446	<0.0001	0.0002	0.0002	<0.0001	0.0995	0.0085
Dose of GAA		0.2799	0.3149	0.3284	0.0027	0.1238	0.2359	0.0349	0.4695	0.3473
Linear		0.6864	0.3658	0.2865	0.2617	0.0274	0.0401	0.3775	0.0877	0.0471
Quadratic		0.7868	0.0772	0.9835	0.2192	0.9049	0.6801	0.2637	0.6572	0.5152
Alfalfa hay × dose of GAA		<0.0001	0.1989	0.0748	0.0012	0.0274	0.0011	0.0021	0.0438	0.0121

^1^ *b* is the asymptotic CH_4_ production (mL CH_4_ g^−1^ DM); *c* is the rate of CH_4_ production (mL CH_4_ h^−1^); *Lag* is the initial delay before CH_4_ production begins (h). ^2^ SEM, standard error of the mean.

**Table 4 animals-13-01719-t004:** Parameters and in vitro ruminal production of carbon monoxide (CO) in response to different percentages of alfalfa (*Medicago sativa* L.) hay and doses of guanidinoacetic acid (GAA) in the diet after 4, 24 and 48 h of fermentation.

Alfalfa Hay (%)	Dose of GAA (g g^−1^ Diet)	CO Production					
		Parameters ^1^			mL CO g^−1^ DM Incubated		
		*b*	*c*	*Lag*	4 h	24 h	48 h
10	0.0000	3.0867	0.0614	5.14	0.0294	0.6530	3.2352
	0.0005	2.3867	0.0053	6.40	0.0132	0.4976	3.5657
	0.0010	4.4267	0.0419	5.78	0.0212	0.7865	4.2513
	0.0015	5.2767	0.0740	4.42	0.0333	0.9961	4.5540
	0.0020	5.4600	0.1362	2.63	0.0349	1.0831	5.4524
	0.0025	4.5767	0.1192	2.71	0.0415	0.9328	4.5694
	0.0030	4.5400	0.1531	3.27	0.0437	1.1410	7.9702
	SEM ^2^	0.68045	0.02497	1.316	0.00557	0.16812	0.96506
	Dose of GAA	0.0574	0.0086	0.3129	0.0186	0.1414	0.0617
	Linear	0.0391	0.7259	0.7035	0.6255	0.1709	0.3503
	Quadratic	0.7731	0.4133	0.5444	0.1585	0.8559	0.7673
25	0.0000	7.7433	0.1620	3.40	0.0190	0.8815	7.8135
	0.0005	8.6800	0.1744	3.44	0.0223	0.9532	8.8230
	0.0010	8.2267	0.1695	3.28	0.0218	0.9709	8.3586
	0.0015	7.1367	0.1606	3.54	0.0224	0.8366	7.1847
	0.0020	6.1867	0.1432	2.92	0.0240	1.0288	6.2225
	0.0025	8.2133	0.1648	3.39	0.0221	1.0196	8.2938
	0.0030	7.5400	0.1196	5.16	0.0213	0.5214	7.2556
	SEM ^2^	1.01042	0.02428	0.534	0.00301	0.10047	1.09523
	Dose of GAA	0.6725	0.7221	0.1691	0.9517	0.0397	0.6872
	Linear	0.6776	0.9681	0.8557	0.4421	0.7567	0.6909
	Quadratic	0.5352	0.7884	0.7835	0.7596	0.3790	0.5320
100	0.0000	5.8233	0.1175	4.96	0.0061	0.3159	5.6202
	0.0005	5.8967	0.1698	3.52	0.0049	0.5169	5.9736
	0.0010	6.4300	0.1098	4.75	0.0049	0.4480	5.0598
	0.0015	4.4233	0.1016	4.49	0.0091	0.4945	3.9633
	0.0020	6.3633	0.1697	3.65	0.0079	0.6257	6.4153
	0.0025	4.5633	0.0854	4.48	0.0073	0.5894	3.7541
	0.0030	5.5267	0.0517	5.46	0.0049	0.3702	3.1422
	SEM ^2^	0.79748	0.04234	1.275	0.00123	0.09110	0.65469
	Dose of GAA	0.4550	0.4457	0.9291	0.1537	0.2480	0.0227
	Linear	0.2349	0.7936	0.7982	0.1094	0.1874	0.0952
	Quadratic	0.2023	0.9962	0.9891	0.0959	0.7073	0.7431
Pooled SEM ^2^		0.84062	0.03165	1.102	0.00372	0.12471	0.92367
*p*-value							
Alfalfa hay		<0.0001	0.0005	0.2818	<0.0001	<0.0001	<0.0001
Linear		<0.0001	0.0001	0.2109	<0.0001	0.7950	<0.0001
Quadratic		0.3249	0.7190	0.3237	<0.0001	<0.0001	0.0020
Doses of GAA		0.9153	0.6899	0.4968	0.0084	0.0617	0.8625
Linear		0.9295	0.9516	0.6984	0.2668	0.1260	0.6713
Quadratic		0.1969	0.7968	0.7212	0.1461	0.6620	0.4530
Alfalfa hay × dose of GAA		0.1093	0.0287	0.5601	0.0077	0.0413	0.0109

^1^ *b* is the asymptotic CO production (ppm CO g^−1^ DM); *c* is the rate of CO production (ppm CO h^−1^); *Lag* is the initial delay before CO production begins (h). ^2^ SEM, standard error of the mean.

**Table 5 animals-13-01719-t005:** Parameters and in vitro ruminal production of hydrogen sulfide (H_2_S) in response to different percentages of alfalfa (*Medicago sativa* L.) hay and doses of guanidinoacetic acid (GAA) in the diet after 4, 24 and 48 h of incubation.

Alfalfa Hay (%)	Dose of GAA (g g^−1^ Diet)	H_2_S Production					
		Parameters ^1^			mL H_2_S g^−1^ DM Incubated		
		*b*	*c*	*Lag*	4 h	24 h	48 h
10	0.0000	0.1443	0.0001	7.97	0.0016	0.0117	0.0490
	0.0005	0.0186	0.0008	5.73	0.0006	0.0045	0.0278
	0.0010	0.0287	0.0021	6.12	0.0004	0.0049	0.0266
	0.0015	0.1154	0.0347	6.19	0.0002	0.0028	0.0196
	0.0020	0.1349	0.0053	6.68	0.0002	0.0029	0.0260
	0.0025	0.0859	0.0032	5.61	0.0002	0.0031	0.0200
	0.0030	0.3613	0.0006	8.60	0.0003	0.0033	0.0347
	SEM ^2^	0.10255	0.01278	1.121	0.00009	0.00102	0.00591
	Dose of GAA	0.3428	0.4881	0.4326	<0.0001	0.0002	0.0441
	Linear	0.8449	0.0761	0.2805	<0.0001	<0.0001	0.0034
	Quadratic	0.4341	0.3445	0.4960	0.0004	0.0744	0.3035
25	0.0000	0.0143	0.0005	6.05	0.0001	0.0022	0.0267
	0.0005	0.0378	0.0002	6.97	0.0001	0.0022	0.0253
	0.0010	0.0218	0.0002	3.97	0.0002	0.0027	0.0260
	0.0015	0.0259	0.0003	6.83	0.0001	0.0022	0.0206
	0.0020	0.0820	0.0003	6.01	0.0002	0.0028	0.0178
	0.0025	0.0191	0.0004	5.06	0.0001	0.0028	0.0243
	0.0030	0.0393	0.0012	4.51	0.0001	0.0018	0.0236
	SEM ^2^	0.02106	0.00030	1.428	0.00003	0.00038	0.00278
	Dose of GAA	0.3653	0.2516	0.6931	0.7583	0.4321	0.3162
	Linear	0.7027	0.6440	0.7063	0.4577	1.0000	0.1431
	Quadratic	0.9494	0.5941	0.1788	0.2071	0.2721	0.5134
100	0.0000	0.0155	0.0001	5.98	0.0002	0.0016	0.0255
	0.0005	0.0133	0.0006	6.58	0.0001	0.0017	0.0263
	0.0010	0.0023	0.0003	7.35	0.0001	0.0018	0.0213
	0.0015	0.4484	0.0007	7.51	0.0004	0.0024	0.0193
	0.0020	0.0228	0.0001	7.57	0.0005	0.0041	0.0357
	0.0025	0.1446	0.0022	6.65	0.0004	0.0041	0.0403
	0.0030	0.0403	0.0001	8.99	0.0003	0.0029	0.0568
	SEM ^2^	0.17054	0.00061	1.313	0.00009	0.00066	0.00654
	Dose of GAA	0.5217	0.2355	0.7660	0.0536	0.0714	0.0141
	Linear	0.0943	0.5433	0.4238	0.1383	0.4089	0.5116
	Quadratic	0.2902	0.8599	0.7139	0.2463	0.7785	0.8976
Pooled SEM ^2^		0.11553	0.00739	1.294	0.00008	0.00073	0.00534
*p*-value							
Alfalfa hay		0.3172	0.2086	0.0724	<0.0001	<0.0001	0.0136
Linear		0.1408	0.1218	0.1291	<0.0001	<0.0001	0.0560
Quadratic		0.7450	0.3878	0.0811	0.3792	0.0110	0.0217
Doses of GAA		0.4699	0.4671	0.7369	<0.0001	0.0012	0.0034
Linear		0.1495	0.0602	0.8688	<0.0001	<0.0001	0.0026
Quadratic		0.1865	0.3239	0.3077	0.0007	0.1852	0.5690
Alfalfa hay × dose of GAA		0.4531	0.5014	0.6710	<0.0001	<0.0001	0.0032

^1^ *b* is the asymptotic H_2_S production (ppm H_2_S g^−1^ DM); *c* is the rate of H_2_S production (ppm H_2_S h^−1^); *Lag* is the initial delay before H_2_S production begins (h). ^2^ SEM, standard error of the mean.

**Table 6 animals-13-01719-t006:** In vitro rumen fermentation profile and CH_4_ conversion efficiency in response to different percentages of alfalfa (*Medicago sativa* L.) hay and doses of guanidinoacetic acid (GAA) in the diet.

Alfalfa Hay (%)	Dose of GAA (g g^−1^ Diet)	In Vitro Rumen Fermentation Profile ^1^				CH_4_ Conversion Efficiency ^2^		
		pH	DMD	SCFA	ME	CH_4_:SCFA	CH_4_:ME	CH_4_:OM
10	0.0000	6.98	75.58	6.40	7.20	63.83	9.04	15.16
	0.0005	6.89	78.12	5.36	6.67	81.88	10.61	16.51
	0.0010	6.87	79.77	6.12	7.06	165.30	23.73	14.02
	0.0015	6.83	82.21	8.19	8.13	304.18	49.21	13.53
	0.0020	6.87	84.27	8.92	8.50	269.79	45.29	13.87
	0.0025	6.83	84.22	8.48	8.28	174.98	28.63	24.71
	0.0030	7.03	85.35	8.51	8.29	174.97	28.92	55.95
	SEM ^3^	0.055	1.455	0.437	0.225	66.508	10.924	4.750
	Dose of GAA	0.1541	0.0023	0.0002	0.0002	0.1790	0.1343	0.0002
	Linear	0.0747	0.0062	0.0117	0.0116	0.0229	0.0210	0.8118
	Quadratic	0.5789	0.6317	0.0454	0.0454	0.8217	0.6931	0.9562
25	0.0000	7.52	68.43	4.30	6.53	108.19	11.45	17.38
	0.0005	7.50	68.25	4.05	6.40	180.37	18.39	27.40
	0.0010	7.53	60.23	4.08	6.41	136.08	14.08	21.14
	0.0015	7.46	70.51	4.43	6.59	59.00	6.38	9.80
	0.0020	7.45	71.25	4.40	6.58	72.12	7.75	11.84
	0.0025	7.55	69.39	4.17	6.46	180.34	18.74	28.18
	0.0030	7.64	65.48	4.08	6.42	113.14	11.47	17.04
	SEM ^3^	0.014	1.582	0.214	0.109	19.580	2.799	4.750
	Dose of GAA	0.0621	0.2894	0.4031	0.4082	0.0978	0.1108	0.1239
	Linear	0.3398	0.6346	0.5681	0.5699	0.7723	0.7147	0.6655
	Quadratic	0.7094	0.5987	0.6943	0.7055	0.664	0.6708	0.6796
100	0.0000	7.21	81.52	6.73	7.37	116.18	16.99	29.07
	0.0005	7.22	80.57	6.50	7.26	112.92	16.07	26.91
	0.0010	7.22	79.93	6.71	7.37	130.91	19.21	33.00
	0.0015	7.23	80.43	6.90	7.46	124.35	18.46	32.03
	0.0020	7.22	85.23	6.86	7.44	139.08	20.45	35.19
	0.0025	7.26	80.03	7.01	7.52	145.62	21.72	37.85
	0.0030	7.27	80.56	6.37	7.19	199.67	28.42	47.62
	SEM ^3^	0.031	3.214	0.133	0.068	41.413	4.387	6.663
	Dose of GAA	0.0142	0.2924	0.2615	0.2633	0.3059	0.3629	0.3933
	Linear	0.1960	0.6547	0.5010	0.5235	0.4151	0.4278	0.4350
	Quadratic	0.3136	0.0340	0.0987	0.0986	0.3183	0.3523	0.3703
Pooled SEM ^3^		0.037	7.232	0.291	0.149	46.625	6.986	5.462
*p*-value								
Alfalfa hay		<0.0001	<0.0001	<0.0001	<0.0001	0.0892	0.0009	<0.0001
Linear		<0.0001	0.8821	<0.0001	<0.0001	0.1345	0.0447	<0.0001
Quadratic		<0.0001	<0.0001	<0.0001	<0.0001	0.1020	0.0010	0.0007
Dose of GAA		0.0011	0.0151	<0.0001	<0.0001	0.4709	0.2184	0.0002
Linear		0.0447	0.1709	0.0053	0.0054	0.0882	0.0384	0.6436
Quadratic		0.9170	0.0533	0.0149	0.0151	0.6557	0.9323	0.4082
Alfalfa hay × dose of GAA		0.2397	0.1320	<0.0001	<0.0001	0.0438	0.0359	0.0086

^1^ pH is ruminal pH; DMD is dry matter degradability (%); SCFA is short-chain fatty acids (mmol g^−1^ DM) after 24 h of incubation; ME is the metabolizable energy (MJ kg^−1^ DM) after 24 h of incubation; ^2^ CH_4_:SCFA is the methane:short-chain fatty acid ratio (mmol mmol^−1^) after 24 h of incubation; CH_4_:ME is the methane:metabolizable energy ratio (g MJ^−1^) after 24 h of incubation; CH_4_:OM is the methane:organic matter ratio (mL g^−1^). ^3^ SEM, standard error of the mean.

## Data Availability

The data presented in this study are available upon request from the corresponding authors.

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
