# Peer review of "Effect of Dietary Guanidinoacetic Acid Levels on the Mitigation of Greenhouse Gas Production and the Rumen Fermentation Profile of Alfalfa-Based Diets"

_animals, 2023, doi:10.3390/ani13111719_

Round 1

Reviewer 1 Report

Comments to the Author

Emissions of greenhouse gases have a substantial influence on climate change. The management and availability of feed resources, especially for, contribute to reduce the production of enteric methane. According to the manuscript, “Effect of dietary guanidinoacetic acid levels on the mitigation of greenhouse gas production and the rumen fermentation profile of alfalfa-based diets” There are a few errors in the document at various points, thus it has to be revised. My suggestions would be as follows:

1. ABSTRACT: The author has to be improved and this section of the report may be concise.

2. The research's introduction section was nearly completely re-concisely explained by an emphasis on the GAA review literature.

3. In the section on materials and procedures, I have noticed:

- Why did the author choose to use the AH percentage at 10, 25, and 100% without using a linear portion? How does the polynomial interpretation respond to this?

- Why didn’t you design the isonitrogenous experimental diet?

- There is not a linear time interval for the total gas production, which was assessed at 2, 4, 12, 24, 27, and 48 hours of incubation.

4. Result and discussion section is required fundamental reconsideration.

- Why did the author designate the parameter in Table 1 as DM-1 digested when the kinetics of the production of gases were examined using the DM basis? Meanwhile, DM incubation was used to characterize CH4, CO, and H2S.

- I have several questions regarding the statistical findings in Table, particularly the interactions between the AH and GAA factors. Why do some numbers exhibit significance for each of those factors but not for their interaction?

5. In conclusion, the lowest CH4 conversion efficiency per OM was in 10%AH, yet the author chose to highlight 25%AH as the best. Do I have this right, or do we need to increase CH4 production?

6. The references, please double-check the formatting of several journal abbreviations, including Journal of Animal Feed Science and Technology, as well as other formatting following the journal's instructions.

Author Response

RESPONSE LETTER- REVIEWER 1

Manuscript ID:

Animals-2343813

Title:

“Effect of dietary guanidinoacetic acid levels on the mitigation of greenhouse gas production and the rumen fermentation profile of alfalfa-based diets"

Journal

Animals

Section:

Animal Nutrition

Special Issue:

Nutritional Strategies to Control Enteric Methane Production of Ruminants

Dear Assigned Reviewers,

We appreciate the review of our manuscript, and the comments you made, which undoubtedly helped us to improve the quality. We have made changes to the manuscript document based on your suggestions and using "change control", and below is a transcript of your suggestions in black and our responses to each suggestion in blue.

Sincerely,

The Author's

______________________________________________________________________

Emissions of greenhouse gases have a substantial influence on climate change. The management and availability of feed resources, especially for, contribute to reduce the production of enteric methane. According to the manuscript, “Effect of dietary guanidinoacetic acid levels on the mitigation of greenhouse gas production and the rumen fermentation profile of alfalfa-based diets” There are a few errors in the document at various points, thus it has to be revised. My suggestions would be as follows:

  1. ABSTRACT: The author has to be improved and this section of the report may be concise.

Response: We have modified the summary to improve comprehension.

  1. The research's introduction section was nearly completely re-concisely explained by an emphasis on the GAA review literature.

Response: We appreciate your comment.

  1. In the section on materials and procedures, I have noticed:

- Why did the author choose to use the AH percentage at 10, 25, and 100% without using a linear portion? How does the polynomial interpretation respond to this?

Response: We used these percentages of alfalfa simulating that they were diets for livestock producing meat (10 % AH), milk (25 % AH) and in grazing conditions (100 % AH). Considering that the AH levels are unequally spaced we use the ORPOL function in PROC IML of SAS to obtain the appropriate coefficients for the contrast statement.

- Why didn’t you design the isonitrogenous experimental diet?

Response: We used these percentages of alfalfa simulating that they were diets for livestock producing meat (10 % AH), milk (25 % AH) and in grazing conditions (100 % AH).

- There is not a linear time interval for the total gas production, which was assessed at 2, 4, 12, 24, 27, and 48 hours of incubation.

Response: Thanks for the observation, but even if there is not a lineal time interval is possible to estimate kinetics of production of gas.

  1. Result and discussion section is required fundamental reconsideration.

- Why did the author designate the parameter in Table 1 as DM-1 digested when the kinetics of the production of gases were examined using the DM basis? Meanwhile, DM incubation was used to characterize CH4, CO, and H2S.

Response:  In this Table we express the parameters in both DM-1 and DM degraded-1. This last unit of expression is very interesting because we can observe the production of gas directly from the dry matter that was used by the microbiota.

- I have several questions regarding the statistical findings in Table, particularly the interactions between the AH and GAA factors. Why do some numbers exhibit significance for each of those factors but not for their interaction?

Response: In these cases, the effect of each factor was not dependent on one another, that is, regardless of the percentage of alfalfa hay, the effect of GAA could be the same.

  1. In conclusion, the lowest CH4 conversion efficiency per OM was in 10%AH, yet the author chose to highlight 25%AH as the best. Do I have this right, or do we need to increase CH4 production?

Response: We modified the conclusion for a better understanding.

  1. The references, please double-check the formatting of several journal abbreviations, including Journal of Animal Feed Science and Technology, as well as other formatting following the journal's instructions.

Response: References were corrected.

Reviewer 2 Report

Dear authors,

First of all, I would like to congratulate all the authors for their extraordinary work in this section of “Nutritional Strategies to Control Enteric Methane Production of Ruminants”. You have included all the necessary information in this article. 

The manuscript was well-written and the content was informative and well-presented. I commend the authors for their comprehensive and systematic command of the topic. The manuscript will be a valuable contribution to this journal.

However, I’ve mentioned a few minor comments that need to be addressed before the manuscript can be published. I mentioned some minor corrections which need to be corrected in the comment section of the main manuscript file.

Thank you. 

Author Response

RESPONSE LETTER- REVIEWER 2

Manuscript ID:

Animals-2343813

Title:

“Effect of dietary guanidinoacetic acid levels on the mitigation of greenhouse gas production and the rumen fermentation profile of alfalfa-based diets"

Journal

Animals

Section:

Animal Nutrition

Special Issue:

Nutritional Strategies to Control Enteric Methane Production of Ruminants

Dear Assigned Reviewers,

We appreciate the review of our manuscript, and the comments you made, which undoubtedly helped us to improve the quality. We have made changes to the manuscript document based on your suggestions and using "change control", and below is a transcript of your suggestions in black and our responses to each suggestion in blue.

Sincerely,

The Author's

______________________________________________________________________

Dear authors,

First of all, I would like to congratulate all the authors for their extraordinary work in this section of “Nutritional Strategies to Control Enteric Methane Production of Ruminants”. You have included all the necessary information in this article. 

The manuscript was well-written and the content was informative and well-presented. I commend the authors for their comprehensive and systematic command of the topic. The manuscript will be a valuable contribution to this journal.

However, I’ve mentioned a few minor comments that need to be addressed before the manuscript can be published. I mentioned some minor corrections which need to be corrected in the comment section of the main manuscript file.

Thank you. 

Dear Assigned Reviewer,

We appreciate your positive feedback about our work, and have addressed all of your feedback in the manuscript with track changes enabled, with the exception of the following:

Line 139: Please mention about the SCFA content for the feed analysis also.

Response: In Table 1 we do not place SCFA content because it was not analyzed in the diets.

Lines 592-594: Too old reference (Reference 31). Please try to mention the most recent article to this reference

Response: We appreciate your comment, but we believe that the right thing to do is to give credit to the person who established the methodology and not to those who use it in their studies.

Reviewer 3 Report

Lines 118-120 - Why were these levels of GAA used? Better description of why or how these levels were selected is needed. Was this based on previous research?

Table 1 - How were digestible energy and digestible crude protein determined?

Line 143 - Good to use fluid from multiple animals.

Lines 154-159 - I assumed you used 3 flasks for each treatment. This should be stated. How did you determine thed number of flasks to use forf each treatment?

Lines 164-168 - Is there a referenced that verifies the ability of the Drager device to detect these gases?

Lines 215-229 - This  whicj mesnsaresection (and other similat sections) could be reworked. It is difficult to link the text with the figure. It is also difficult to determine which means are significantly different. As an exsample, is the total gas production in Figure 1 from the 25% AH diet at 48 hours significsntly different from the other 2 treatments? One possibility is to discuss results due to AH level, then results from GAA and then interactions. This would mske it easier for the reader. 

Line 215 - Figures and tables can't show. Change this throughout the paper.

Line 319 - How as ME determined?

Lines 346-361 - This approch for describing results from SH is good snd esy to read. What about GAA levels?

Other descussion sections - Much better in the way presented.

Conclusion - Is there an overall level of AH and GAA that as best? I might be good to add some comments relative to the incubation time snd rumen retention time for lactating cows. A time of 20-30 hours fits lactating cows while 48 fits dry cows. If you do an in vivo evaluation, what level(s) of AH and GAA  would be used?    

Author Response

RESPONSE LETTER- REVIEWER 3

Manuscript ID:

Animals-2343813

Title:

“Effect of dietary guanidinoacetic acid levels on the mitigation of greenhouse gas production and the rumen fermentation profile of alfalfa-based diets"

Journal

Animals

Section:

Animal Nutrition

Special Issue:

Nutritional Strategies to Control Enteric Methane Production of Ruminants

Dear Assigned Reviewers,

We appreciate the review of our manuscript, and the comments you made, which undoubtedly helped us to improve the quality. We have made changes to the manuscript document based on your suggestions and using "change control", and below is a transcript of your suggestions in black and our responses to each suggestion in blue.

Sincerely,

The Author's

______________________________________________________________________

Lines 118-120 - Why were these levels of GAA used? Better description of why or how these levels were selected is needed. Was this based on previous research?

Response: These levels of GAA were based in previous studies in beef cattle (Li et al., 2021; Li et al., 2022; Liu et al., 2021) and sheep (Chao et al., 2019). We made the description in the text.

Table 1 - How were digestible energy and digestible crude protein determined?

Response: These data were estimate of tabular data so were removed of the table.

Line 143 - Good to use fluid from multiple animals.

Response: It's right.

Lines 154-159 - I assumed you used 3 flasks for each treatment. This should be stated. How did you determine thed number of flasks to use forf each treatment?

Response: Statistically, the minimum number of replicates is three, so we use three replicates per treatment.

Lines 164-168 - Is there a referenced that verifies the ability of the Drager device to detect these gases?

Response: The technical specifications of the equipment can be found on the manufacturer's website, and we have already proven its reliability in previous studies, which can be found online (https://doi.org/10.1016/j.jevs.2022.104021; https://doi.org/10.1016/j.jevs.2022.104049).

Lines 215-229 - This  whicj mesnsaresection (and other similat sections) could be reworked. It is difficult to link the text with the figure. It is also difficult to determine which means are significantly different. As an exsample, is the total gas production in Figure 1 from the 25% AH diet at 48 hours significsntly different from the other 2 treatments? One possibility is to discuss results due to AH level, then results from GAA and then interactions. This would mske it easier for the reader.

Response: We made the relevant corrections.

Line 215 - Figures and tables can't show. Change this throughout the paper.

Response: We made the relevant corrections.

Line 319 - How as ME determined?

Response: The ME was calculated with the formula two described in the materials and methods section.

Lines 346-361 - This approch for describing results from SH is good snd esy to read. What about GAA levels?

Other descussion sections - Much better in the way presented.

Response: We have modified the paragraph.

Conclusion - Is there an overall level of AH and GAA that as best? I might be good to add some comments relative to the incubation time snd rumen retention time for lactating cows. A time of 20-30 hours fits lactating cows while 48 fits dry cows. If you do an in vivo evaluation, what level(s) of AH and GAA  would be used?    

Response: We have modified the conclusion and added some recommendations.

Round 2

Reviewer 1 Report

Comments to the Author

Based on the previous suggestions, your manuscript might be reconsidered only one point.

-        Why did the author conclude that a diet containing 10% AH had the greatest benefits when administered at doses equivalent to or higher because they enhanced CH4?

Author Response

RESPONSE LETTER- REVIEWER - 1

Manuscript ID:

Animals-2343813

Title:

“Effect of dietary guanidinoacetic acid levels on the mitigation of greenhouse gas production and the rumen fermentation profile of alfalfa-based diets"

Journal

Animals

Section:

Animal Nutrition

Special Issue:

Nutritional Strategies to Control Enteric Methane Production of Ruminants

Dear Assigned Reviewer,

We have made changes to the manuscript document based on your suggestions and using "change control", and below is a transcript of your suggestions in black and our responses to each suggestion in blue.

Sincerely,

The Author's

Based on the previous suggestions, your manuscript might be reconsidered only one point.

Why did the author conclude that a diet containing 10% AH had the greatest benefits when administered at doses equivalent to or higher because they enhanced CH4?

Author answer:

The diet with 10 % HA was the only one that presented a significant effect due to the addition of GAA, and from the evaluated doses we assert that 0.0015 g g-1 DM DM diet or more are better because they increased DMD, SCFA and ME. By increasing these values with the mentioned doses, it is assumed that the response of the animals will increase positively in comparison with the rest of the doses, and for this reason we suggest carrying out an in vivo evaluation to validate our results. We hope we have resolved your question.

Reviewer 3 Report

Thanks for making the changes.

Author Response

RESPONSE LETTER- Reviewer 3

Manuscript ID:

Animals-2343813

Title:

“Effect of dietary guanidinoacetic acid levels on the mitigation of greenhouse gas production and the rumen fermentation profile of alfalfa-based diets"

Journal

Animals

Section:

Animal Nutrition

Special Issue:

Nutritional Strategies to Control Enteric Methane Production of Ruminants

Dear Assigned Reviewer,

We have made changes to the manuscript document based on your suggestions and using "change control", and below is a transcript of your suggestions in black and our responses to each suggestion in blue.

Sincerely,

The Author's

Thanks for making the changes.

Author answer:

We appreciate the review of our manuscript and your comments.
